# Context-Scaling versus Task-Scaling in In-Context Learning

## Abstract

Transformers exhibit In-Context Learning (ICL), a phenomenon in which these models solve new tasks by using examples in the prompt without additional training. In our work, we analyze two key components of ICL: (1) context-scaling, where model performance improves as the number of in-context examples increases and (2) task-scaling, where model performance improves as the number of pre-training tasks increases. While transformers are capable of both context-scaling and task-scaling, we empirically show that standard Multi-Layer Perceptrons (MLPs) with vectorized input are only capable of task-scaling. To understand how transformers are capable of context-scaling, we first propose a significantly simplified transformer that performs ICL comparably to the original GPT-2 model in statistical learning tasks (e.g., linear regression, teacher-student settings). By analyzing a single layer of our proposed model, we identify classes of feature maps that enable context scaling. Theoretically, these feature maps can implement the Hilbert estimate, a model that is provably consistent for context-scaling. We then show that using the output of the Hilbert estimate along with vectorized input empirically enables both context-scaling and task-scaling with MLPs. Overall, our findings provide insights into the fundamental mechanisms of how transformers are able to learn in context.

## 1 Introduction

Pre-trained large language models exhibit In-Context Learning (ICL) capabilities, allowing them to adapt to new tasks based exclusively on input context without updating the underlying model parameters (Brown et al., 2020).

| Input (Prompt) | Output | Task |
|---|---|---|
| $(1, 2, 3), (4, 5, 9), (10, -9, 1), (5, 6, ?)$ | 11 | In each triplet $(a, b, c)$: $c = a + b$ |
| $(4, 3, 1), (9, 0, 9), (10, 8, 2), (17, 17, ?)$ | 0 | In each triplet $(a, b, c)$: $c = a - b$ |
| $(1, 2, 5), (2, 3, 8), (3, 4, 11), (5, 6, ?)$ | 17 | In each triplet $(a, b, c)$: $c = a + 2b$ |
| $(2, 1, 7), (3, 4, 18), (5, 2, 16), (4, 3, ?)$ | 17 | In each triplet $(a, b, c)$: $c = 2a + 3b$ |

As an example of ICL, suppose we prompt a language model using a sequence of $N$ triples of numbers in which each of the first $N - 1$ triples follows a given pattern and the goal is to fill in the missing number in triple $N$. In the table above, we give an example where $N = 4$. A model is capable of filling in the missing entry by using the in-context examples present in the prompt without seeing these specific examples in the training data. What makes in-context learning possible?

As a step toward answering this question, recent research analyzed a family of ICL problems where, for example, the task data was generated using linear regression, student-teacher neural networks, and decision trees (see Garg et al., 2022; Akyürek et al., 2022; Bai et al., 2023; Ahn et al., 2023; Zhang et al., 2024a; Raventós et al., 2023; Wu et al., 2024, for an non-exhaustive list of examples). In these problems, a transformer was first pre-trained on $T$ tasks where the data in each task was generated from a given family of functions. For example, each task may involve predicting the last element in a tuple as a linear combination of the other elements, as was shown in the table above.

The pre-trained transformer was then tested on $N$ samples from a new task that is drawn from the same family but was not seen during pretraining. Such a setup allows for understanding the effect of various factors including neural network architecture, the number of pre-training tasks $T$, and context length $N$ on ICL.

Thus far, the ability of models to learn in context has been broadly defined as their ability to generalize to unseen tasks based on context examples without updating the model parameters. We observe that there are two settings for studying generalization in ICL. The first, which we call *context-scaling*, refers to the ability of the model to improve as the context length $N$ increases while the number of pre-training tasks $T$ is fixed. The second, *task-scaling* refers to the ability of a model to improve as $T$ increases while $N$ is fixed.

**(A) Task-scaling**     **(B) Context-scaling**

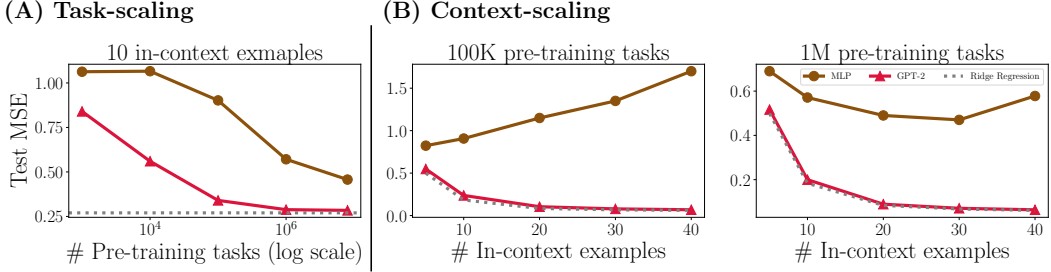

Figure 1: Task-scaling and context-scaling of GPT-2 architecture transformers versus MLPs for ICL with linear regression tasks. **(A)** Task-scaling abilities of these models with 10 in-context examples. **(B)** Context-scaling abilities of these models with $10^5$ (left) and $10^6$ (right) pre-training tasks. Experimental details are provided in Appendix A.

It is a priori unclear whether a model capable of context-scaling is also capable of task-scaling and vice-versa. For example, as we show in Figure 1A, both transformers and standard Multi-Layer Perceptrons (MLPs) are capable of task-scaling on ICL of linear regression tasks. In contrast, only transformers benefit from an increasing number of context examples as shown in Figure 1B.

*What mechanism enables transformers, but not MLPs, to context-scale?*

To identify such a mechanism, we begin by constructing a bare-bones transformer with all key, query, and value weight matrices fixed to be the identity matrix. We refer to our simplified model as Simplified GPT (SGPT). Despite its simplicity, we find that SGPT is competitive with GPT-2 architecture transformers (Radford et al., 2019) for a variety of ICL problems including linear regression, student-teacher networks, decision trees, and sparse linear regression (Garg et al., 2022).

Upon further analyzing a one-layer version of SGPT, we find that SGPT applies a feature map to data that enables context-scaling. To illustrate how such a feature map can be effective for context-scaling, consider the following input data for ICL:

$$A = \begin{bmatrix} x_1 & x_2 & \cdots & x_{N-1} & x_N \\ y_1 & y_2 & \cdots & y_{N-1} & 0 \end{bmatrix}^\top \in \mathbb{R}^{N \times (d+1)} ; \tag{1}$$

where $x_i \in \mathbb{R}^d$ and our goal is to predict $y_N$. We show that SGPT first applies a feature map $\psi : \mathbb{R}^{N \times (d+1)} \to \mathbb{R}^{N \times (d+1)}$ to $A$ and then trains an MLP on the last row of $\psi(A)$, denoted $\psi(A)_{N,:}$.

By varying $\psi$, we show that the scalar $\psi(A)_{N,d+1}$ itself is an effective estimate for $y_N$. For example, when $\psi(A) = (AA^\top)A$, $\psi(A)_{N,d+1}$ can implement one-step GD with a scalar stepsize for linear regression (using context examples $(x_i, y_i)_{i=1}^{N-1}$) (Von Oswald et al., 2023). When the feature follows an isotropic Gaussian distribution, this estimator is consistent (as both the numbers of pre-training tasks and context examples grow) and nearly matches the optimally tuned ridge regression (Mahankali et al., 2024; Wu et al., 2024). More generally, we show that $\psi(A)_{N,d+1}$ can implement the Hilbert estimate (Devroye et al., 1998), which provides a statistically consistent estimate for general families of tasks beyond linear regression as the context length $N$ approaches infinity. As such, our results provably establish that a one-layer transformer is capable of context-scaling for any family of tasks.

Further, we show that these features can enhance the capabilities of MLPs for context-scaling. In particular, by concatenating features from $\psi(A)_{N,:}$ with the vectorized input, $A_v \in \mathbb{R}^{N(d+1)}$, we enable task-scaling and context-scaling simultaneously in MLPs, which previously could only task-scale when trained on $A_v$.

We summarize our findings as follows:

- We show that a simplified transformer, SGPT, with all key, query, and value weight matrices fixed to the identity matrix, is competitive with GPT-2 on a range of ICL tasks.
- We analyze a one-layer version of SGPT, demonstrating that this model is capable of context-scaling solely through the use of a feature map applied to input data.
- We theoretically show that this feature map can be modified to perform kernel smoothing to impute the missing element in each task using the other in-context examples. Upon selecting the Hilbert estimate as the choice of smoother (Devroye et al., 1998), this imputed element becomes a statistically optimal (consistent) estimate as the context length approaches infinity.
- We show that by providing MLPs with a concatenation of features from our proposed feature map and the vectorized input for each task, the model can simultaneously achieve both context-scaling and task-scaling.

Overall, we identify key mechanisms that transformers can use to provably generalize to new, unseen tasks given a large context.

## 2 PRIOR WORK

**ICL in controlled settings.** The work by Garg et al. (2022) initiated the study of ICL in statistical learning tasks, such as linear regression, decision tree learning, and teacher-student neural network learning. They showed that transformers such as GPT-2 (Radford et al., 2019), when pre-trained with a huge number of independent tasks (around $3.2 \times 10^7$ independent tasks, each presented once), can exhibit a strong ICL ability, in the sense that during inference time, the performance of pre-trained transformers matches the Bayes optimal or the best-known algorithms for these tasks. These results were later extended to other settings (see Akyürek et al., 2022; Raventós et al., 2023; Bai et al., 2023; Li et al., 2023; Tong & Pehlevan, 2024, for examples). In particular, Raventós et al. (2023) empirically showed transformers can achieve nearly Bayes optimal ICL even when pre-trained with multiple passes over a much smaller number of independent tasks, and Bai et al. (2023) showed by construction that transformers can select optimal algorithms in context if the task prior is a mixture of distributions. These works together suggest that transformers can achieve both task-scaling and context-scaling for ICL. On the other hand, Tong & Pehlevan (2024) empirically showed that pretrained MLPs can achieve ICL when the context length is fixed and is the same during pre-training and inference. However, it was unclear whether MLPs can achieve context-scaling, and our work empirically gives a negative answer. Motivated by this, we study the mechanism inside transformers that enable both task and context scaling.

**Linear attention and one-step gradient descent.** The theory of ICL is best understood in linear regression with a fixed context length (therefore only concerning the task-scaling). Specifically, Von Oswald et al. (2023) showed by construction that single linear attention can implement one-step gradient descent (GD) in context. Ahn et al. (2023); Zhang et al. (2024a); Mahankali et al. (2024); Zhang et al. (2024b) proved that optimally pre-trained single linear attention is equivalent to a one-step GD. Wu et al. (2024) proved that one-step GD nearly matches the Bayes optimal algorithm for inference and can be pre-trained efficiently with finite independent tasks. Later works such as Cheng et al. (2024) connected nonlinear attention to functional one-step GD in feature space. These works together have substantially furthered our understanding of single linear attention for ICL of linear regression with a fixed context length. However, these results assume (or are only sharp the context length is fixed, and in particular, they fail to explain the observed context-scaling ability of pre-trained transformers.

**Softmax attention and kernel smoothers.** The connection between softmax attention and kernel smoothers was first pointed out by Tsai et al. (2019). Specifically, by setting the query and key

matrices to be the same, an attention component can be viewed as a kernel-smoother (that uses a learnable semi-positive definite kernel). Empirical evidence suggests that using shared a matrix for query and key matrices does not significantly impact the performance of transformers (Tsai et al., 2019; Yu et al., 2024). Later, theoretical works Chen et al. (2024); Collins et al. (2024) utilized this connection to study the ICL of softmax attention in both linear (Chen et al., 2024) and nonlinear regression tasks (Collins et al., 2024). In these settings, softmax attention achieves ICL by implementing a kernel smoother with an optimal bandwidth parameter (achieved by training query and key matrices). Compared with these works, we demonstrate that transformers can achieve ICL using an attention component that does not contain trainable parameters (that is, setting query, key, and value matrices to identity matrices). Our results suggest that the transformer architecture is capable of performing ICL without needing to explicitly learn any hyperparameters in the attention head. We explain this by connecting to a hyper-parameter-free yet statistically consistent kernel smoother given by the Hilbert estimate (Devroye et al., 1998).

**Approximation ability of transformers.** The transformer is known to be a versatile architecture that can implement efficient algorithms (by forwarding passing an input prompt) in many scenarios (see Akyürek et al., 2022; Bai et al., 2023; Guo et al., 2023; Lin et al., 2023; Gatmiry et al., 2024, for a non-exhaustive list of examples). These constructive results, while explaining ICL from an approximation theory perspective, heavily exploit the trainability of query, key, and value weight matrices. In this work, we show that transformers are competitive with GPT-2 architecture transformers for ICL even when all of their query, key, and value weight matrices are fixed to be the identity matrix. Our results suggest the prior constructive results may not fully explain the ICL ability of transformers.

## 3 PRELIMINARIES

In this section, we outline the problem setup, training details, architectural details, and mathematical preliminaries for our work.

**Problem formulation.** For all ICL tasks studied in our work, we consider T pre-training tasks, each with input data of the form:

$$A_t = \begin{bmatrix} x_1^t & x_2^t & \cdots & x_N^t \\ y_1^t & y_2^t & \cdots & y_N^t \end{bmatrix}^\top \in \mathbb{R}^{(N) \times (d+1)}, \tag{2}$$

where $t = 1, \ldots, T$ indexes the tasks, $N$ denotes the maximum context length, and $d$ denotes the input data dimension. We define the loss function $L(\theta)$ as:

$$L(\theta; A_1, \ldots, A_T) := \frac{1}{T} \sum_{t=1}^{T} \left[ \frac{1}{N} \sum_{i=1}^{N} (M_\theta(A_t^i) - y_{i+1}^t)^2 \right],$$

where

$$A_t^i = \begin{bmatrix} x_1^t & x_2^t & \cdots & x_i^t & x_{i+1}^t \\ y_1^t & y_2^t & \cdots & y_i^t & 0 \end{bmatrix}^\top \in \mathbb{R}^{(i+1) \times (d+1)},$$

and $M_\theta(\cdot)$ denotes the model with trainable parameters $\theta$. The tasks $A_t$ are uniformly sampled from the family of tasks $\mathcal{F}$ (e.g., linear regression with a Gaussian prior), representing the distribution of tasks relevant to the in-context learning problem.

**Attention.** Given three matrices $A_1, A_2, A_3 \in \mathbb{R}^{N \times m}$, attention layers implement functions $g : \mathbb{R}^{N \times m} \times \mathbb{R}^{N \times m} \times \mathbb{R}^{N \times m} \to \mathbb{R}^{N \times m}$ defined as follows,

$$g(A_1, A_2, A_3) := \phi \left( \frac{1}{\sqrt{m}} A_1 A_2^\top \right) A_3; \tag{3}$$

where $\phi : \mathbb{R}^{N \times N} \to \mathbb{R}^{N \times N}$ is a generic function that could be a row-wise softmax function (Vaswani et al., 2017), an entry-wise activation function such as ReLU, or just an identify map. For self-attention layers, we are typically given one input matrix $A \in \mathbb{R}^{N \times m}$ and three weight matrices $W_Q, W_K, W_V \in \mathbb{R}^{m \times m}$. In this case, the matrices $A_1, A_2, A_3$ are computed respectively as $AW_Q, AW_K$, and $AW_V$.

**Kernel function.** Kernel functions are positive-semidefinite functions that map pairs of inputs to real values (Schölkopf et al., 2002). Formally, given $x, y \in \mathbb{R}^d$, a kernel $K : \mathbb{R}^d \times \mathbb{R}^d \to \mathbb{R}$ is a function of the form $K(x, y) = \langle \psi(x), \psi(y) \rangle_{\mathcal{H}}$, where $\psi : \mathbb{R}^d \to \mathcal{H}$ is referred to as a feature map from $\mathbb{R}^d$ to a Hilbert space $\mathcal{H}$. For matrix inputs $A \in \mathbb{R}^{m \times d}$ and $B \in \mathbb{R}^{n \times d}$, we let $K(A, B) \in \mathbb{R}^{m \times n}$ such that $K(A, B)_{ij} = K(A_i, B_j)$ where $A_i$ and $B_j$ denote the $i$-th and $j$-th rows of $A$ and $B$ respectively.

**Kernel smoother.** Given a kernel $K : \mathbb{R}^d \times \mathbb{R}^d \to \mathbb{R}$ and a set of points $(\mathbf{x}_i, y_i)_{i=1}^n$ where $\mathbf{x}_i \in \mathbb{R}^d$ and $y_i \in \mathbb{R}$, the kernel smoother estimate at point $\mathbf{x}$ is a function of the form

$$\hat{f}_{K,n}(\mathbf{x}) := \frac{\sum_{i=1}^n K(\mathbf{x}, \mathbf{x}_i) y_i}{\sum_{i=1}^n K(\mathbf{x}, \mathbf{x}_i)} . \tag{4}$$

We will reference kernel smoothers in Section 5.

**Hilbert estimate.** The Hilbert estimate is a kernel smoother using the kernel

$$H(x, x') := \frac{1}{\|x - x'\|_2^d}, \tag{5}$$

where $\|\cdot\|_2$ denotes the $\ell_2$-norm in $\mathbb{R}^d$. The key property of the Hilbert estimate that we use is that it is a consistent (asymptotically optimal) estimate. In particular, at almost all $x$, as the number of samples $n$ goes to infinity, $\hat{f}_{H,n} \to f^*$ in probability where $f^*(x) = \mathbb{E}[y|X = x]$ denotes the Bayes optimal predictor (Devroye et al., 1998).

## 4 SIMPLIFIED TRANSFORMER MODEL PERFORMS IN CONTEXT LEARNING

In this section, we introduce our simplified transformer model, SGPT, and demonstrate that it is competitive with GPT-2-type architectures on various ICL tasks. To construct SGPT, we fix all key, query, and value weights to be the identity matrix in the GPT-2 architecture. Consequently, the attention mechanism (defined in equation 3) reduces to the function:

$$g(H) := \phi(HH^\top)H \in \mathbb{R}^{N \times (d+1)}. \tag{6}$$

We define $\phi$ to be a function that performs row-wise $\ell_1$ normalization on its argument. To further simplify our model, we remove the final linear layer of each MLP block (i.e., our MLP blocks have one linear layer), along with batch normalization and the skip connection after the MLP layer. These details are further outlined in Appendix A.

We consider the following ICL tasks from prior work: (1) linear regression with a single noise level (Akyürek et al., 2022), (2) linear regression with multiple noise levels used in (Bai et al., 2023), (3) sparse linear functions used in (Garg et al., 2022), (4) two-layer ReLU neural networks (Garg et al., 2022), and (5) decision trees (Garg et al., 2022). Below, we explain the problem setup and state our results for each of these five synthetic tasks.

**Linear regression with fixed noise.** The problem setting is as follows:

$$x \in \mathbb{R}^d \sim \mathcal{N}(0, I_d), \quad y = \beta^\top x + \epsilon \text{ with } \beta \sim \mathcal{N}\left(0, \frac{I_d}{d}\right), \ \epsilon \sim \mathcal{N}(0, \sigma^2).$$

In this setting, prior work by Bai et al. (2023), showed that on all context lengths, GPT-2 architecture transformers can perform comparably to task-specific, optimally-tuned ridge regression. In Figure 2, we provide evidence that SGPT matches the performance of these GPT-2 models.

**Linear regression with multi noise level.** The problem setting is as follows:

$$x \in \mathbb{R}^d \sim \mathcal{N}(0, I_d), \quad y_i = \beta^\top x_i + \epsilon$$

$$\text{with } \beta \sim \mathcal{N}\left(0, \frac{I_d}{d}\right) \text{ and } \epsilon \sim \begin{cases} \mathcal{N}(0, \sigma_1^2), & \text{with probability } \frac{1}{2} \\ \mathcal{N}(0, \sigma_2^2), & \text{with probability } \frac{1}{2} \end{cases}.$$

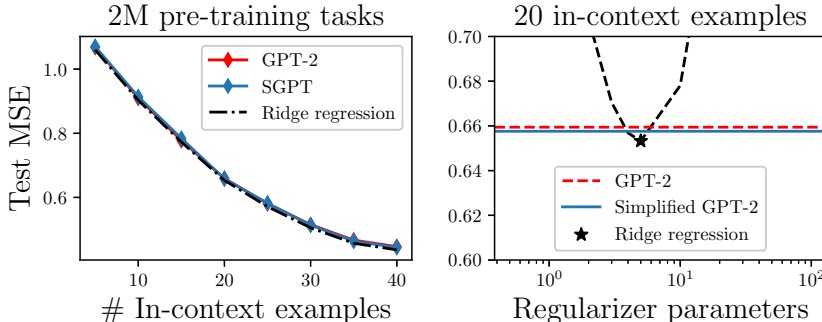

Figure 2: Linear regression with a single noise level. **Left panel.** Performance across varying context lengths (context-scaling). **Right panel.** Effect of regularization on performance for a fixed number of in-context examples. Experimental details are given in Appendix A.

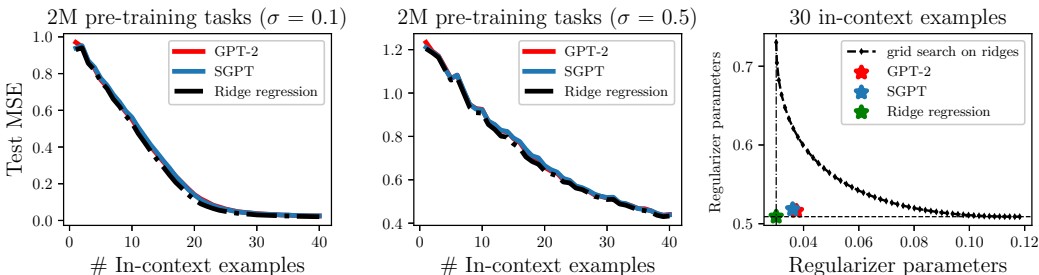

Figure 3: Linear regression with multiple noise levels. **Left and middle panels:** Performance across varying context lengths (context-scaling). **Right panel:** Effect of regularization on performance for a fixed number of in-context examples. Experimental details are given in Appendix A.

In this setting, prior work by Bai et al. (2023) demonstrated that GPT-2 architecture transformers can achieve performance comparable to that of task-specific, optimally tuned ridge regression across all context lengths and for both noise levels. They refer to the model's ability to adapt to the noise level as *algorithm selection*. In Figure 3, We demonstrate that SGPT performs comparably to GPT-2 architecture transformers in this setting, exhibiting similar algorithm selection capabilities.

**Two-layer ReLU Neural Networks.** Following the work of Garg et al. (2022), we consider the following nonlinear problem setting where data for each task are generated using two-layer neural networks. In particular, data are generated according to

$$x \in \mathbb{R}^d \sim \mathcal{N}(0, I_d), \quad y = \sum_{j=1}^{r} \alpha_j \phi(w_j^\top x);$$

where $\alpha_j, w_j$ are randomly initialized parameters of a fixed two-layer neural network, $\phi$ denotes the element-wise ReLU activation function, and $r = 100, d = 20$ (as selected in prior work). Garg et al. (2022) demonstrated that GPT-2 architecture transformers can match the performance of student networks (i.e., networks of the same architecture initialized differently and trained using Adam optimizer Kingma (2014)). In Figure 4(B), we show that SGPT can match the performance of GPT-2 architecture transformers on this task.

**Decision Tree.** Following the work of Garg et al. (2022), we consider a nonlinear problem setting where data for each task are generated using depth-four trees. For a task corresponding to a tree $f$, we have:

$$x \sim \mathcal{N}(0, I_d), \quad y = f(x), \tag{7}$$

Previously, Garg et al. (2022) demonstrated that GPT-2 architecture transformers can perform in-context learning on this family of non-linear functions, outperforming XGBoost as a baseline. In

Figure 4(A), we show that SGPT is also capable of in-context learning (ICL) in this setting, performing comparably to GPT-2 architecture and similarly outperforming XGBoost. We trained XGBoost models using the same hyperparameters as in (Garg et al., 2022).

**Sparse linear functions.** Following the work by Garg et al. (2022), we consider the class sparse linear regression problems. In this setting, data are generated according to

$$x \in \mathbb{R}^d \sim \mathcal{N}(0, I_d), \quad y = \beta^\top x, \tag{8}$$

where $\beta \sim \mathcal{N}(0, I_d)$ and we zero out all but $s$ coordinates of $\beta$ uniformly at random for each task. As in prior work, we select $d = 20, s = 3$. In Figure 4C , we demonstrate that SGPT is capable of ICL for this class of functions, performing comparably to GPT-2 architecture transformers and closely to the Lasso estimator (Tibshirani, 1996), while significantly outperforming the ordinary least square (OLS) baseline.

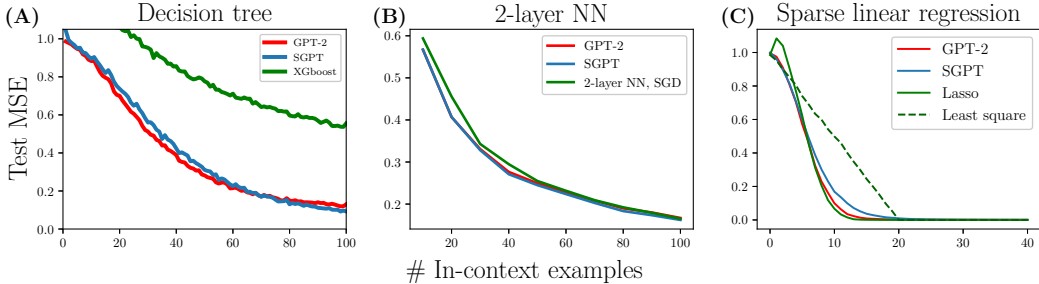

Figure 4: Nonlinear ICL tasks. Context-scaling capability of SGPT versus GPT-2 architecture transformers when trained on 2 million pre-training tasks. In all cases, the errors are normalized so that the trivial zero predictor achieves an error of 1.* Experimental details are given in Appendix A.

## 5 KERNEL SMOOTHING CAN ACHIEVE CONTEXT-SCALING

We begin this section by demonstrating that even one layer of SGPT is capable of context scaling. In particular, in Figure 5, we train a one-layer model on five different context lengths and test on the same lengths for the tasks considered in the previous section. In all four problem settings, it is evident that using more context improves performance. Below, we analyze this simplified one layer model in order to pinpoint how it is capable of context scaling.

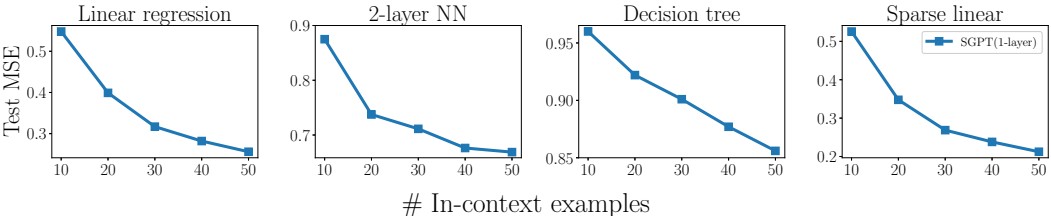

Figure 5: Context-scaling with one-layer SGPT. Experimental details are provided in Appendix A.

In particular, the model we analyze is identical to one layer of SGPT up to the omission of remaining skip connections (for more details, see Appendix A). The model implements a function $f : \mathbb{R}^{N \times (d+1)} \to \mathbb{R}$ and takes the form below:

$$f(A) := \left[\sigma\left(\psi(A) W^{(1)}\right) W^{(2)}\right]_N = \sigma\left(\psi(A)_{N,:} W^{(1)}\right) W^{(2)}; \tag{9}$$

---

*For decision trees, we found that GPT-2 performs poorly when using the input structure of concatenating $x$ and $y$. Therefore, we used the pre-trained model fromGarg et al. (2022)

where $\psi : \mathbb{R}^{N \times (d+1)} \to \mathbb{R}^{N \times (d+1)}$ is a feature map (generalizing the attention function defined in equation 6), $A \in \mathbb{R}^{(N) \times (d+1)}$ denotes the input data, and $k$ denotes the embedding dimension with $W^{(1)} \in \mathbb{R}^{d+1 \times k}, W^{(2)} \in \mathbb{R}^{k \times 1}$.

## 5.1 FEATURE MAP THAT ENABLES CONTEXT-SCALING

The key aspect distinguishing the model in equation 9 from a standard MLP operating on vectorized inputs $A \in \mathbb{R}^{N(d+1)}$ is the feature map $\psi$. As such, we analyze how the feature map $\psi$ transforms an input

$$A = \begin{bmatrix} x_1 & x_2 & \cdots & x_{N-1} & x_N \\ y_1 & y_2 & \cdots & y_{N-1} & 0 \end{bmatrix}^\top \in \mathbb{R}^{N \times (d+1)}.$$

First, we note that upon varying the function $\psi$, the bottom-right element of $\psi(A)$, denoted $\psi(A)_{N,d+1}$, is capable of implementing several well-known estimators, which we describe below. To ease notation, we let $\mathbf{X} := [x_1, x_2, \cdots, x_N]^\top \in \mathbb{R}^{N \times d}$ and $\mathbf{y} := [y_1, \cdots, y_N]^\top$. Detailed derivations of the explicit forms for $\psi(A)_{N,d+1}$ below are presented in Appendix B.

**(1) 1-step GD estimate.** Let $\psi_L(A) := (AA^\top)A$. Then,

$$\psi_L(A)_{N,d+1} = x_N^\top \mathbf{X}^\top \mathbf{y}. \tag{10}$$

Thus, $\psi_L$ computes the estimate arising from a linear predictor trained for one step of gradient descent on the data $(\mathbf{X}, \mathbf{y})$. This estimate has been previously considered as a mechanism through which transformers performed ICL, but there have been no theoretical guarantees for this approach for general ICL tasks beyond linear regression (Von Oswald et al., 2023; Ahn et al., 2023; Zhang et al., 2024a; Mahankali et al., 2024; Zhang et al., 2024b).

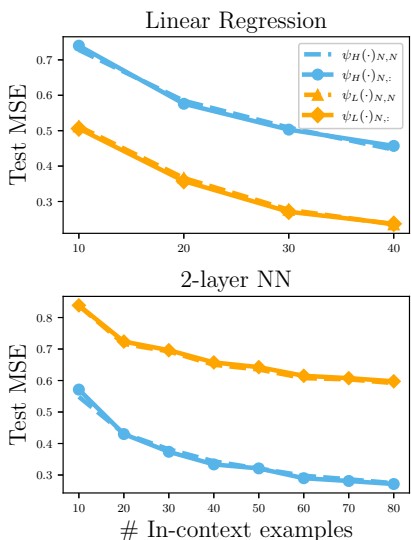

Figure 6: Comparison between using a row of features given by $\psi_K(\cdot)_{N,:}$ and using the scaler given by $\psi_K(\cdot)_{N,d+1}$ for $K \in \{L, H\}$. Here, the feature maps $\psi_L, \psi_H$ are defined in equation 10 and equation 5 respectively.

**(2) Kernel smoother.** Given a kernel $K$, let $\psi_K(A) = \hat{K}(\mathbf{X}, \mathbf{X})A$, where

$$\hat{K}(\mathbf{X}, \mathbf{X})_{i,j} = \begin{cases} \frac{K(x_i, x_j)}{\sum_{j \neq i} K(x_i, x_j)} & \text{if } i \neq j \\ 0 & \text{if } i = j \end{cases}.$$

In this case, $\psi_K(A)_{N,:}$ has the following form

$$\psi_K(A)_{N,:} = \underbrace{\left[ \frac{\sum_{i=1}^{N-1} K(x_N, x_i) x_i^\top}{\sum_{i=1}^{N-1} K(x_N, x_i)} \right.}_{\text{smoothed } d\text{-dimensional features}}, \underbrace{\left. \frac{\sum_{i=1}^{N-1} K(x_N, x_i) y_i}{\sum_{i=1}^{N-1} K(x_N, x_i)} \right]}_{\text{smoothed estimate}} \in \mathbb{R}^{d+1} \tag{11}$$

and the last element $\psi_K(A)_{N,d+1}$ is the kernel smoother estimate,

$$\psi_K(A)_{N,d+1} = \frac{\sum_{i=1}^{N-1} K(x_N, x_i) y_i}{\sum_{i=1}^{N-1} K(x_N, x_i)}. \tag{12}$$

Below, we provide key examples of kernel smoothers that can be implemented by equation 12 upon changing the kernel $K$.

(1) When $K$ is the exponential kernel, i.e., $K(z, z') = e^{-z^\top z}$, then $\psi_K$ implements softmax attention, and $\psi_K(A)_{N,d+1}$ is the kernel smoother corresponding to the exponential kernel.

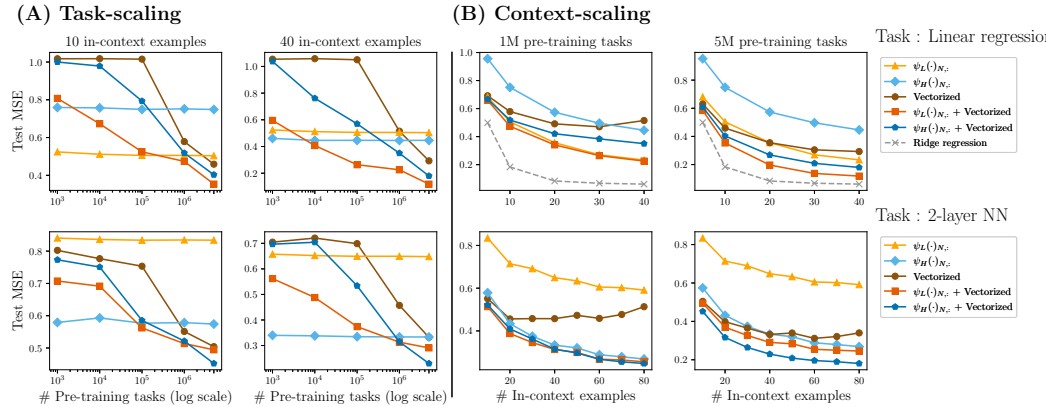

Figure 7: Comparison of MLPs trained using (1) vectorized inputs; (2) features from $\psi_K$ for $K \in L, H$ defined in equation 10 and equation 5; (3) both features from $\psi_K$ and vectorized inputs. We compare performance across two ICL tasks: linear regression and two-layer teacher-student neural networks. **(A)** Task-scaling ability of MLPs using different inputs. **(B)** Context-scaling ability of MLPs using different inputs. MLPs trained on both vectorized inputs and features from $\psi_K$ are able to simultaneously context-scale and task-scale. Experimental details are provided in Appendix A.

(2) When using the kernel $H$ defined in equation 5, then, $\psi_H(A)_{N,d+1}$ implements the Hilbert estimate, which is consistent as the number of in-context examples goes to infinity (Devroye et al., 1998).

In our experiments in Figure 5, we trained an MLP on features computed using $\psi_K(A)_{N,:}$. Yet, the results above suggest that the scalar $\psi_K(A)_{N,d+1}$ alone should be sufficient for context scaling. Indeed, in the case of the Hilbert estimate, this entry alone provides a consistent estimate as the context length goes to infinity. To this end, in Figure 6, we compare the performance of two MLPs when the number of tasks is fixed and the context length increases. The first MLP is trained using $\psi_K(A)_{N,:} \in \mathbb{R}^{d+1}$, and the second is trained on only $\psi_K(A)_{N,d+1}$. The results in this figure confirm that using $\psi_K(A)_{N,d+1}$ is as good as using $\psi_K(A)_{N,:}$ for context-scaling.

## 5.2 Training MLPs that simultaneously context-scale and task-scale

As the Hilbert estimate provides a consistent estimate, our results above show that transformers provably generalize to unseen tasks, when the context length approaches infinity. Nevertheless, the issue with using the Hilbert estimate alone is that the Hilbert estimate is only computed using examples provided in a context. As such, it cannot task-scale unlike MLPs trained on vectorized inputs. We now show that training MLPs on vectorized inputs concatenated with features estimated using $\psi_K(A)$ result in MLPs that can both context-scale and task-scale.

Namely, we revisit the experiment presented in Figure 1 and extend our analysis by training MLPs on three distinct input configurations: (1) vectorized input data; (2) features from $\psi_K(A)_{N,:}$; and (3) the concatenation of vectorized input data and features from $\psi_K(A)_{N,:}$. In our experiments, we consider the feature maps $\psi_L$ and $\psi_H$ discussed in the previous section. The results of training these MLPs is presented in Figure 7 and we summarize the results below.

1. **MLPs with vectorized input data:** Figure 7A demonstrates that these MLPs exhibit task-scaling. Yet, Figure 7B reveals that these MLPs fail to context-scale and performance can even deteriorate with increased context length.

2. **MLPs with features from $\psi_K(\cdot)_{N,:}$:** Figure 7A illustrates that these MLPs do not task-scale, as performance does not improve with an increasing number of tasks. This behavior matches intuition as the Hilbert smoother and 1-step gradient descent features are task-specific and do not leverage inter-task relationships. Yet, Figure 7B shows that these MLPs successfully context-scale, which happens provably for the particular case of $\psi_H$.

3. **MLPs with both vectorized inputs and features from $\psi_K(\cdot)_{N,:}$:** Figure 7A demonstrates that these MLPs are capable of task-scaling, consistent with the performance of MLPs on vectorized data alone. Moreover, in Figure 7B, we now observe that these MLPs are now capable of context-scaling, consistent with the performance of MLPs using the features from $\psi_K(\cdot)_{N,:}$ alone.

These results underscore the importance of the feature map $\psi_K$ for context-scaling and highlight the effectiveness of using both vectorized inputs and features from $\psi_K$ in improving the ability of models to learn in-context.

## 6 CONCLUSION AND LIMITATIONS

**Summary.** In this work, we observed that transformers, unlike MLPs, are able to simultaneously context-scale (improve performance as the context length increases) and task-scale (improve performance as the number of pre-training tasks increases). To better understand this property of transformers, we first identified a simplified transformer (SGPT) that could solve ICL tasks competitively with GPT-2 architecture transformers despite having no trainable key, query, and value weights in attention layers. By studying a one-layer version of SGPT, we identified that the attention operator of SGPT applied a feature map, $\psi$, on input data that enabled context-scaling. In particular, we showed that this feature map could implement kernel smoothers such as the Hilbert estimate, which is a statistically consistent estimator as the context length goes to infinity. As such, our work provably demonstrates that transformers can context-scale, generalizing to new, unseen tasks when provided a large context. We demonstrated the effectiveness of the feature map, $\psi$, for context-scaling by showing that MLPs trained on both features from $\psi$ and vectorized inputs could simultaneously context-scale and task-scale.

**Future work and limitations.** While we have provably established that one-layer transformers can context-scale, we empirically observe that one-layer transformers are not as sample-efficient as deep transformers for both context-scaling and task-scaling. Thus, an important future direction is understanding how depth improves the sample complexity of transformers in both context-scaling and task-scaling settings. Exploring this aspect remains a promising avenue for future research and could provide a comprehensive understanding of ICL, and more broadly, a better understanding of how transformers are able to generalize to new tasks when provided large contexts.

**Reproducibility Statement.** The codebase is included as supplementary material, and we will release the GitHub repository upon acceptance.

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

## A  EXPERIMENTS DETAILS

We provide all experimental details below.

**Problem formulation.**  For all ICL tasks studied in our work, we consider T pretraining tasks, each with input data of the form:

$$X_t = \begin{bmatrix} x_1^t & x_2^t & \cdots & x_N^t \\ y_1^t & y_2^t & \cdots & y_N^t \end{bmatrix}^\top \in \mathbb{R}^{(N)\times(d+1)}$$

where $t \in 1, \ldots, T$ indexes the tasks, $N$ denotes the maximum context length, and $d$ denotes the input data dimension. We define the loss function $L(\theta)$ as:

$$L(\theta; X_1, \ldots, X_T) := \frac{1}{T} \sum_{t=1}^{T} \left[ \frac{1}{N} \sum_{i=1}^{N} (M_\theta(X_t^i) - y_{i+1}^t)^2 \right]$$

where:

$$X_t^i = \begin{bmatrix} x_1^t & x_2^t & \cdots & x_i^t & x_{i+1}^t \\ y_1^t & y_2^t & \cdots & y_i^t & 0 \end{bmatrix}^\top \in \mathbb{R}^{(i+1)\times(d+1)}$$

$M$ denotes the model with trainable parameters $\theta$.

**Vectorized input.**  By vectorizing input, we mean flattening the input into a vector. After vectorization, $X_t^i$ defined above becomes,

$$X_t^i = \begin{bmatrix} x_1^{t\top} & x_2^{t\top} & \cdots & x_i^{t\top} & x_{i+1}^{t\top} & y_1^t & y_2^t & \cdots & y_i^t & 0 \end{bmatrix}^\top \in \mathbb{R}^{(i+1)(d+1)}.$$

**GPT-2.**  We used the GPT-2 implementation from prior work (Garg et al., 2022; Bai et al., 2023), which is based on the Hugging Face implementation (Wolf et al., 2020). Following the approach in these prior works, we modified the embedding layer with a learnable linear layer that maps from the ambient dimension to an embedding dimension.

**SGPT.** To construct SGPT, we make the following modifications to the GPT-2 architecture: (1) we fix all key, query, value weights to the identity; (2) we eliminate all batch-normalization layers; and (3) we remove the second linear layer from each MLP. Following prior work (Garg et al., 2022; Bai et al., 2023), we modified the embedding layer with a linear layer that maps from the ambient dimension to an embedding dimension. In SGPT, this linear layer is not trainable and serves as a fixed random map. We outline the architecture below.

Let $A \in \mathbb{R}^{N \times (d+1)}$ be the input of the model. We initialize a random matrix $W_0 \in \mathbb{R}^{(d+1) \times k}$, where $k$ is the embedding dimension. Defining the input of the $i$-th layer as $H^{(i)}$, we have $H^{(0)} := AW_0$,

$$H^{(i)} = \sigma\Big(\big(g(H^{(i-1)})W_{proj}^{(i)} + H^{(i-1)}\big)W_{MLP}^{(i)}\Big) + g(H^{(i-1)})W_{proj}^{(i)} + H^{(i-1)} \tag{13}$$

where:

- $\sigma$ is the activation function, chosen to be GeLU,

- $g$ is as defined in Equation 6, $\phi$ is row wise $l1$ normalziation.

- $W_{proj}^{(i)} \in \mathbb{R}^{k \times k}$ is the projection matrix,

- $W_{MLP}^{(i)} \in \mathbb{R}^{k \times k}$ is the MLP weight matrix for the $i$-th layer.

The last layer of the network is a linear layer with weights $W_O \in \mathbb{R}^{k \times 1}$.

**MLP architectures.** In all experiments, we use a standard 2-layer ReLU MLP with a width of 1024 units. Given an input vector $x \in \mathbb{R}^{d_{in}}$, the MLP implements a function $f$ of the form

$$f(x) := \sigma(xW_0)W_1 \tag{14}$$

where $W_0 \in \mathbb{R}^{d_{in} \times 1024}$, $W_1 \in \mathbb{R}^{1024 \times 1}$, and $\sigma$ is the ReLU activation function.

**Zero-padding input for MLP.** In all experiments, we always trained **a single MLP for all context lengths** by zero-padding to the largest context length. For example, if the input data is in $\mathbb{R}^d$ and the largest context length is $N_{max}$, then the input dimension of the MLP is $dN_{max}$.

**Expeirmental details for Figure 1.** In this experiment, we trained an 8-layer GPT-2 model with 8 attention heads and a width of 256. The MLP configuration is the same as that in equation 14. We trained and tested the models on the same set of context lengths: 5, 10, 20, 30, and 40.

**Experimental details for Section 4.** We trained both standard GPT-2 architecture transformers and our proposed SGPT with the following configurations:

- Widths: $\{256, 512, 1024\}$,

- Number of layers: $\{2, 4, 6, 8\}$,

- Number of attention heads for GPT-2: $\{2, 4, 8\}$.

The best performance was achieved with an 8-layer model with a width of 256. For the original GPT-2, the optimal configuration used 8 attention heads.

We outline per-task observations and configurations below:

1. **Linear regression with single noise level:** Following the prior work Garg et al. (2022), the input dimension is $d = 20$ and noise level is $\sigma = 0.5$. We trained on context lengths from 10 to 40 with a step size of 5.

2. **Linear regression with two noise levels:** Following the prior work of Garg et al. (2022), the input dimension is $d = 20$ and noise levels are $\sigma_1 = 0.1, \sigma_2 = 0.5$. We trained on context lengths from 1 to 40 with a step size of 1.

3. **Decision tree:** Following the prior work of Garg et al. (2022), the input dimension is $d = 20$ with a tree depth of 4. We note that with the input structure equation 2, GPT-2 performs poorly, so in our figure we used the pretrained model from Garg et al. (2022).

4. **Two-layer ReLU Neural Networks.** As mentioned before, we chose the width of this family of neural networks to be $r = 100$ and the input dimension to be $d = 20$. We trained on context lengths from 1 to 100 with a step size of 10. We observed that unlike our model, GPT-2 does not generalize well for context lengths that it has not been trained on.

5. **Sparse Linear Regression** As mentioned previously, the ambient dimension of the input is $d = 20$, consistent with prior work(Garg et al., 2022), and the effective dimension is $s = 3$. We used `scikit-learn` Pedregosa et al. (2011) for the Lasso and Ordinary least sauare performances.

**Experimental details for Figure 5.** In all tasks, we used input dimension $d = 8$ following the setting in Tong & Pehlevan (2024). We trained and tested both models on context lengths of 10, 20, 30, 40, and 50.

**Experimental details for Figure 6.** We train an MLP (equation 14) on features extracted using $\psi_K(\cdot)_{N,:}$ and linear regression on the scalar $\psi_K(\cdot)_{N,d+1}$.

**Experimental details for Figure 7:**

- **Linear regression:** We used the same settings as used for the model in equation 3 with $d = 8$ and $\sigma = 0.22$. We trained and tested models on the context lengths 5, 10, 20, 30, and 40.

- **2-layer NN task:** We used the same settings as used for the model in 4 with $d = 8, r = 100$. Trained and tested models on the context lengths 10, 20, 30, 40, 50, 60, 70, and 80.

**Hardware.** We used machines equipped with NVIDIA A100 and A40 GPUs, featuring V-RAM capacities of 40GB. These machines also included 8 cores of Intel(R) Xeon(R) Gold 6248 CPU @ 2.50GHz with up to 150 GB of RAM. For all our experiments, we never used more than one GPU, and no model was trained for more than two days.

## B  FEATURE MAP DERIVATION

Let $A = \begin{bmatrix} x_1^\top & y_1 \\ x_2^\top & y_2 \\ \vdots & \vdots \\ x_{N-1}^\top & y_{N-1} \\ x_N^\top & 0 \end{bmatrix} \in \mathbb{R}^{N \times (d+1)}$.

**1-step of GD.**  In this case, we have

$\psi_L(A) = (AA^\top)A$

$= \begin{bmatrix} x_1^\top & y_1 \\ x_2^\top & y_2 \\ \vdots & \vdots \\ x_{N-1}^\top & y_{N-1} \\ x_N^\top & 0 \end{bmatrix} \begin{bmatrix} x_1 & x_2 & \cdots & x_{N-1} & x_N \\ y_1 & y_2 & \cdots & y_{N-1} & 0 \end{bmatrix} A$

$= \begin{bmatrix} x_1^\top x_1 + y_1 y_1 & \cdots & x_1^\top x_{N-1} + y_1 y_{N-1} & x_1^\top x_N \\ x_2^\top x_1 + y_2 y_1 & \cdots & x_2^\top x_{N-1} + y_2 y_{N-1} & x_2^\top x_N \\ \vdots & \cdots & \vdots & \vdots \\ x_{N-1}^\top x_1 + y_{N-1} y_1 & \cdots & x_{N-1}^\top x_{N-1} + y_{N-1} y_{N-1} & x_{N-1}^\top x_N \\ x_N^\top x_1 & \cdots & x_N^\top x_N & x_N^\top x_N \end{bmatrix} \begin{bmatrix} x_1 & y_1 \\ x_2 & y_2 \\ \vdots & \vdots \\ x_{N-1} & y_{N-1} \\ x_N & 0 \end{bmatrix}.$

Thus, $\psi_L(A)_{N,d+1} = x_N^\top \mathbf{X}^\top \mathbf{y}$, where $\mathbf{X} := \begin{bmatrix} x_1^\top \\ \vdots \\ x_N^\top \end{bmatrix} \in \mathbb{R}^{N \times (d+1)}$ and $\mathbf{y} := \begin{bmatrix} y_1 \\ \vdots \\ y_N \end{bmatrix}$. This value is equivalent to the prediction given by using one-step of gradient descent to solve linear regression.

**Kernel smoothers.** In this case, we have

$$\psi_K(A) = \hat{K}(\mathbf{X}, \mathbf{X})A$$

$$= \begin{bmatrix} 0 & \frac{K(x_1,x_2)}{\sum_{i=1 \ i\neq 1}^N K(x_1,x_i)} & \cdots & \frac{K(x_1,x_N)}{\sum_{i=1 \ i\neq 1}^N K(x_1,x_i)} \\ \frac{K(x_2,x_1)}{\sum_{i=1 \ i\neq 2}^N K(x_2,x_i)} & 0 & \cdots & \frac{K(x_2,x_N)}{\sum_{i=1 \ i\neq 2}^N K(x_2,x_i)} \\ \vdots & \vdots & \vdots & \vdots \\ \frac{K(x_N,x_1)}{\sum_{i=1 \ i\neq N}^N K(x_N,x_i)} & \frac{K(x_N,x_2)}{\sum_{i=1 \ i\neq N}^N K(x_N,x_i)} & \cdots & 0 \end{bmatrix} \begin{bmatrix} x_1 & y_1 \\ x_2 & y_2 \\ \vdots & \vdots \\ x_{N-1} & y_{N-1} \\ x_N & 0 \end{bmatrix}.$$

Now last row of the $\psi_K(A)$ is given by

$$\psi_K(A)_{N,:} = \begin{bmatrix} \frac{\sum_{i=1}^{N-1} K(X_N,x_i)x_i}{\sum_{i=1}^{N-1} K(X_N,x_i)} & \frac{\sum_{i=1}^{N-1} K(X_N,x_i)y_i}{\sum_{i=1}^{N-1} K(X_N,x_i)} \end{bmatrix} \in \mathbb{R}^{d+1}.$$

Thus, $\psi_K(A)_{N,d+1}$ is equvialent to the prediction for $x_N$ given by using a kernel smoother with kernel $K$.