# OpenReview forum: "Context-Scaling versus Task-Scaling in In-Context Learning"
_ICLR.cc/2025/Conference — Submitted to ICLR 2025_

### Official Review · Reviewer_DfAQ · 2024-10-28

**Soundness:** 2
**Presentation:** 2
**Contribution:** 1
**Rating:** 3
**Confidence:** 4

**Summary:**

In recent years, there has been a growing interest in the community to understand and build better in-context learning capabilities. In this work, the authors study in-context learning through the lens of two separate abilities: context scaling and task scaling. Context scaling refers to the capability to improve predictions as the number of in-context examples grow while keeping the number of tasks fixed. Task scaling refers to the capability to improve predictions as the number of task grows while keeping the number of in-context examples fixed. The authors show that a lot of in-context learning capabilities arise from the behavior of attention as a kernel smoother. To show this, the authors consider a simplified version of the GPT architecture, which they call SGPT, where the attention blocks are not trained and key, query, and value matrices are set to identity. The authors compare GPT-2 like architecture with SGPT and show that on synthetic in-context   learning tasks from the literature, the performance of the two models match. Further, the authors study context scaling and task scaling in MLPs. First, they show that MLPs with vectorized inputs are capable of task scaling and not capable of context scaling. They show that MLPs with featurized inputs are capable of context scaling but not of task scaling. Finally, they combine the two types of inputs to show MLPs exhibit both context scaling and task scaling.

**Strengths:**

1. I like the lens that the authors use in the paper, i.e., separating in-context learning into task scaling and context scaling. Even though this lens is not completely new, its a useful one.

2. The authors propose a simplification of GPT which is referred to as SGPT. This simplification is a useful one from two perspectives -- it gives way to a model that can be understood theoretically and it possibly indicates that we could simplify the architectures for in-context learning.

3. Overall, the paper is easy to read and reasonably well presented.

**Weaknesses:**

I have several concerns with the paper that I highlight below.

1. **On feature maps for context-scaling:**
a) If you construct a Kernel based mapping of the form in equation 11, then of course due to sheer consistency argument one can say that the map in equation 12 converges to the true function. This would require infinite examples in-context. The whole point of in-context learning is to learn in-context as quickly as possible. From an asymptotics point of view, all the consistent estimators in equation 4 can learn the function. Thus the experiments you conduct in Figure 5 are not really interesting. If you took the model in equation 4 itself, then so long as the Kernel you use gives a consistent estimate, one could argue that as sequence length n increases in equation 4 the performance will improve and this model will context scale. Would you elaborate, what is really the important and new insight one should take from Figure 5?

    b) In the first part of Section 5.1, the authors construct a mapping which is based on linear attention and show the equivalence to one step of gradient descent. This result is already known from previous works such as Von Oswald's work.  Can the authors explain what is truly new in Section 5.1?


2. **On the results with simplified transformer:** As i stated in the strengths section, I found the result with simplified transformer intriguing. However, there are few important ablations that would have helped understand this section better. First of all, the authors change the attention to a linear attention l1 normalization operation. What would have happened if the authors only used linear attention and no normalization? While the model is close to standard GPT in terms of performance, how much of this is due to learning that happens due to the MLPs at different depths? Put differently, can the authors show the impact of depth on the performance, i.e., as depth increases model learns more complex hierarchy of features that allow it to match GPT? If that is the case, then this should be clearly stated in the paper too, that depth was crucial to match the performance of GPT at short context lengths. In some sense, if depth is crucial to match the performance, then the role of kernel smoothing alone is not a crucial one.

3. **On Section 5.2**: In this section, the authors study various variants of MLPs to study context scaling and task scaling capabilities. I find somethings unclear here.


   a) Firstly, I find the fact that vectorized MLPs with sufficient capacity not able to context scale not clear. The training data takes the form P, x, where P is the prompt that contains (x,y) pairs from the task of interest, x is the current query. In the case, where the model class is unrestricted, then the learner should ideally learn E[y| P,x], where y is the true label and expectation is over the distribution of prompts and query, label pairs. If the task of interest is linear regression (studied in Garg et al.), the coefficients of the regression are drawn from an isotropic Gaussian, and the features are drawn from an isotropic Gaussian as well, then
   E[y| P,x] = ((X.t X + sigma I)^{-1} X.t Y).t x, where sigma is noise. This is solution to standard ridge regression. Provided the model class be it transformer or be it MLP is sufficently expressive and contains ((X.t X + sigma I)^{-1} X.t Y).t x, then in principle MLP should also be capable of context scaling. So my question to the authors is "If MLP has sufficient capacity, what stops it from implementing a ridge regression solution at different context lengths up to the maximum context length determined by size of vectorized input?" Basically, if the MLP is not able to learn, then it has to be an argument that is not explained by expressivity but by learnability. Perhaps the optimization cannot find the global minimum in the above case easily?

     b) Secondly, the authors show that MLPs with features from kernel smoothes can context scale. I don't quite get what's the surprise here. Isn't the input feature to these MLPs itself guaranteed to be consistent in infinite context limit?


4. **On the role of context-length:** The authors state that existing works fail to explain the context scaling capabilities of transformers.  In the example, I gave in 3a), the Bayes optimal predictor is implemented by the transformer. This Bayes optimal predictor of course improves with context length. In this sense, I don't quite get why do the authors say that existing works fail to explain context scaling capabilities.

5. **On MLPs and context scaling**: The authors also mention that it is unclear whether MLPs scale with context and they are the first to dive into this. Perhaps the authors have missed https://arxiv.org/pdf/2311.18194.

Overall, I don't feel I learned something insightful from the paper, the only thing I liked was the experiment with SGPT.

**Questions:**

In the weakness section, I provide both the concerns and the questions for the authors.

---

> ### Author Response · Authors · 2024-11-18
> **Official Comment by Authors (Part 1)**
>
> We thank the reviewer for the comments. We will address the concerns as following,
>
> > **Reviewer comment:** On feature maps for context-scaling
> >a) If you construct a Kernel based mapping of the form in equation 11, ... Thus the experiments you conduct in Figure 5 are not really interesting. .... Would you elaborate, what is really the important and new insight one should take from Figure 5?
>
> **Our response :** The point of Figure 5 is to set up the question of context scaling.  While it is true that Hilbert kernels can context scale, it is not a priori clear that SGPT can context-scale since SGPT is not a kernel smoother.  Furthermore, prior to our work, it was not clear what part of context-scaling was due to the attention  alone and what part was due to key, query, and value weights.
>
> > **Reviewer comment:** b) In the first part of Section 5.1, the authors construct a mapping which is based on linear attention and show the equivalence to one step of gradient descent. This result is already known from previous works such as Von Oswald's work. Can the authors explain what is truly new in Section 5.1?
>
> **Our response :** In Section 5.1, we show that attention can implement the Hilbert estimate, which is statistically consistent for any statistical in context learning task.  In contrast one step of gradient descent is not generally statistically consistent for statistical in context learning tasks. Thus, implementing 1-step of gradient descent is not a major point. We simply show that it can be obtained from our framework as a special case (note that we cite  Oswald et al. (2023)).
>
> > **Reviewer comment:** On the results with simplified transformer: As i stated in the strengths section, I found the result with simplified transformer intriguing. ... What would have happened if the authors only used linear attention and no normalization? While the model is close to standard GPT in terms of performance, how much of this is due to learning that happens due to the MLPs at different depths? Put differently, can the authors show the impact of depth on the performance, i.e., as depth increases model learns more complex hierarchy of features that allow it to match GPT? If that is the case, then this should be clearly stated in the paper too, that depth was crucial to match the performance of GPT at short context lengths. In some sense, if depth is crucial to match the performance, then the role of kernel smoothing alone is not a crucial one.
>
>
> **Our response :**
> 1. Role of L1 Normalization: We found the l1 normalization to be helpful in avoiding numerical instability when training multi-layer SGPT.
> 2. Impact of Depth on Performance: In all of our results, we used the same depth for SGPT as GPT2. See Appendix A for details on how we selected depth for these models.
>
> > **Reviewer comment:** On Section 5.2: In this section, the authors study various variants of MLPs to study context scaling and task scaling capabilities. I find somethings unclear here.
> > a) Firstly, I find the fact that vectorized MLPs with sufficient capacity not able to context scale not clear. ... So my question to the authors is "If MLP has sufficient capacity, what stops it from implementing a ridge regression solution at different context lengths up to the maximum context length determined by size of vectorized input?" Basically, if the MLP is not able to learn, then it has to be an argument that is not explained by expressivity but by learnability. Perhaps the optimization cannot find the global minimum in the above case easily?
>
> **Our response :**  Our work is unrelated to expressivity arguments and instead, focuses on learnability.  We show that MLPs, on their own, do not learn a solution that generalizes to large context length even though in theory they are capable of doing so.  In case we misunderstood your question, we would appreciate a clarification.
>
> > **Reviewer comment:** b) Secondly, the authors show that MLPs with features from kernel smoothes can context scale. I don't quite get what's the surprise here. Isn't the input feature to these MLPs itself guaranteed to be consistent in infinite context limit?
>
> **Our response :** While kernel smoothers can context-scale, they cannot task-scale. MLPs with the combination of raw data and kernel features can perform both.

---

> ### Author Response · Authors · 2024-11-18
> **Official Comment by Authors (Part 2)**
>
> > **Reviewer comment:**  On the role of context-length: The authors state that existing works fail to explain the context scaling capabilities of transformers. In the example, I gave in 3a), the Bayes optimal predictor is implemented by the transformer. This Bayes optimal predictor of course improves with context length. In this sense, I don't quite get why do the authors say that existing works fail to explain context scaling capabilities.
>
> **Our response :** Again, while there exist constructions of neural networks that can implement optimal predictors, it is far from clear that these models can learn these solutions from data.  This is a primary limitation of the works which focus on constructions, as noted in our related works section.
>
> > **Reviewer comment:**  On MLPs and context scaling: The authors also mention that it is unclear whether MLPs scale with context and they are the first to dive into this. Perhaps the authors have missed https://arxiv.org/pdf/2311.18194.
>
> **Our response :** We are not sure what part of the linked paper is relevant to our work.  In particular the models they considered are DeepSet which are not standard MLPs operating on vectorized data.  We would appreciate a clarification.
>
> We hope our responses address your concerns. If any points remain unclear, we’d be happy to clarify. If the manuscript now aligns with your expectations, we’d appreciate your consideration of an updated score.

---

> > ### Comment · Reviewer_DfAQ · 2024-11-26
> >
> > I thank the authors for their responses. I appreciate the clarifications. However, even after reading your responses, I do not think there was some important insights that I take away from this work. I would appreciate if the authors try to sharpen the message of the paper, do some better experimentation to draw more insightful conclusions. For instance, the experiment with SGPT involves a multi-layer architecture. Even though query, key, and value matrices are set to identity, MLP matrices are not and they should play a role in improving in-context learning too. Hence, it is not clear how much does attention structure only contribute to in-context learning.

---

### Official Review · Reviewer_NaCy · 2024-10-29

**Soundness:** 2
**Presentation:** 2
**Contribution:** 3
**Rating:** 6
**Confidence:** 3

**Summary:**

This study examines task versus context scaling in various in-context regression tasks, finding that Transformers (but not MLPs) can scale with increasing context sizes, as well as tasks. The authors suggest that a Transformer's ability to context-scale stems from its ability to implement kernel smoothers in the attention matrix. Equipping an MLP with features derived from these kernel smoothers enable it to also context-scale.

**Strengths:**

I found the paper overall to be very well-written and a pleasure to read. The topic is extremely important, particularly in our post-ChatGPT era, and deals with a critical ability in Transformers. The contrast to MLPs sparks a fascinating discussion about the relative merits of different architectures.

**Weaknesses:**

I would love to see this manuscript published at ICLR, but there are a few oversights that prevent me from assigning a higher score. If these are able to be addressed, I will be delighted to raise my score.

The discussion on context-scaling in MLPs appears to be drawing from prior work by Tong and Pehlevan (https://arxiv.org/abs/2405.15618). The authors claim that MLPs do not context-scale, but Tong and Pehlevan seem to be showing otherwise. I may be misunderstanding both sides here, but Fig 1d and 1i of Tong and Pehlevan appear to demonstrate that at least MLP-Mixers continue to do well for arbitrary contexts. While MLP performance decays as context length increases in ICL regression, it doesn't appear to be the case for ICL classification. Were you able to look at classification tasks as well? Further, it looks like Tong and Pehlevan made the choice of plotting *excess* MSE above Bayes optimal rather than raw MSE. Because Bayes optimal MSE falls as context length increases, zero excess MSE would imply context-scaling. Looking at Figure 6b, for fewer dimensions than the $d= 8$ you tested, it does appear that MLPs adhere to the Bayes optimal level before failing at longer contexts.

Overall, it would appear that context-scaling in MLPs does happen, but is bottlenecked by some aspect of insufficient data and long inputs, rather than some inability to implement kernel smoothers. MLP-Mixers, which do not have any product interactions that could implement a kernel smoother in an obvious way, continue to do well also. Indeed, it looks like in your Figure 7B, you do demonstrate some context-scaling in unmodified MLPs for 5M-pretraining tasks (top row), quite similar to using $\psi_L$ features.

Additionally, I thought that kernel smoothers are weak in high dimensions, and require a dataset size that is exponential in the input dimension in order to interpolate well -- a classical curse of dimensionality. However, modern Transformers routinely handle token embeddings with dimensions that number in the tens of thousands, which would presumably defeat a kernel smoother even if it were exposed to an Internet-scale corpus -- and in-context, no less! It's quite possible I misunderstand this aspect of your analysis, but it seems implausible that a kernel smoother interpretation of attention is applicable to real-world Transformers?

Finally, you mention there is a theoretical connection between having identity KQV matrices and the Hilbert estimate kernel, but I couldn't find the derivation in your manuscript. This could very well be a blatant oversight on my part, but could you point me to its location?

**Questions:**

See weaknesses above.

---

> ### Author Response · Authors · 2024-11-18
>
> We thank the reviewer for the comments. We will address the concerns as following,
>
> > **Reviewer comment:** The discussion on context-scaling in MLPs appears to be drawing from prior work ....  Were you able to look at classification tasks as well? Further, it looks like Tong and Pehlevan made the choice of plotting excess MSE above Bayes optimal rather than raw MSE. Because Bayes optimal MSE falls as context length increases, zero excess MSE would imply context-scaling. Looking at Figure 6b, for fewer dimensions than the d=8 you tested, it does appear that MLPs adhere to the Bayes optimal level before failing at longer contexts.
>
> **Our response :** Per the reviewer’s suggestion, we added an additional experiment showing that MLPs cannot context scale even for classification.  Below, we consider a classification problem where the label is given by the “sign” of the output of a linear regression model - if the sign is positive, the label is 0 and if it is negative, the label is 1.  We find MLP performance first improves and then gets worse with increasing context length indicating a failure to context scale.
>
> | Context Length      | 5       | 10      | 20      | 30      | 40      | 60      | 80      | 100     | 120     |
> |---------------------|---------|---------|---------|---------|---------|---------|---------|---------|---------|
> | 100k Pre-training   | 66.61%  | 68.68%  | 69.21%  | 69.35%  | 68.43%  | 70.16%  | 70.14%  | 69.84%  | 69.69%  |
> | 1M Pre-training     | 69.9%   | 74.25%  | 76.49%  | 77.66%  | 78.08%  | 77.07%  | 77.23%  | 75.39%  | 74.92%  |
>
> Furthermore, Fig. 1d of Tong & Pehlevan does not provide information about context scaling as it is measuring excess risk, which refers to the difference between the model’s MSE and that of optimal ridge regression.  optimal ridge regression varies with context length and thus, it is possible for excess risk to increase even though MSE decreases.  In their figure, even transformers have higher excess risk (error bars indicate performance approaches that of random prediction) as context length increases even though we know these models do context-scale.
>
> > **Reviewer comment:** Overall, it would appear that context-scaling in MLPs does happen, but is bottlenecked by some aspect of insufficient data and long inputs, ... Indeed, it looks like in your Figure 7B, you do demonstrate some context-scaling in unmodified MLPs for 5M-pretraining tasks (top row), quite similar to using \psi_L features.
>
> **Our response :** Our claim is that standard MLPs trained on vectorized data do not context scale.  Note that by context-scaling, we refer to a setting in which the number of pretraining tasks is fixed, and we expect performance to improve as context examples increase – rather than initially improve and then degrade at a certain point.  Even for the 5M pre-training case referenced by the reviewer, increasing the context length results in worse MLP performance, as we show below.  Note that performance continually improves for \psi_L, \psi_H features while MLP performance first improves and then worsens as context length increases:
>
>
> | **Context Length** | **5**   | **10**  | **20**  | **30**   | **40**   | **60**   | **80**  | **100**  | **120**  | **160**  |
> |---------------------|---------|---------|---------|----------|----------|----------|---------|----------|----------|----------|
> | **ψ_L**            | 0.73    | 0.54    | 0.39    | 0.305    | 0.25     | 0.199    | 0.164   | 0.147    | 0.138    | 0.134    |
> | **ψ_H**            | 1.11    | 0.816   | 0.597   | 0.497    | 0.431    | 0.371    | 0.332   | 0.313    | 0.282    | 0.275    |
> | **Vectorized**      | 0.67    | 0.51    | 0.384   | 0.348    | 0.325    | 0.3      | 0.287   | 0.293    | 0.305    | 0.335    |
>
> > **Reviewer comment:** Additionally, I thought that kernel smoothers are weak in high dimensions, ...  It's quite possible I misunderstand this aspect of your analysis, but it seems implausible that a kernel smoother interpretation of attention is applicable to real-world Transformers?
>
> **Our response :** The method we are proposing (one-layer SGPT) is not a kernel smoother, it is the combination of kernel smoother and MLP. We further show in Figure 7, that the combination has better sample complexity than kernel smoother (or one-step of GD) alone.
>
> > **Reviewer comment:** Finally, you mention there is a theoretical connection ... but I couldn't find the derivation in your manuscript. … could you point me to its location?
>
> **Our response :**  The derivation is in Appendix B.
>
>
> We hope our responses and additional experiments address your concerns. If any points remain unclear, we’d be happy to clarify. If the manuscript now aligns with your expectations, we’d appreciate your consideration of an updated score.

---

> > ### Comment · Reviewer_NaCy · 2024-11-21
> >
> > Thanks for the additional details! I have a few remaining questions
> >
> > > Per the reviewer’s suggestion, we added an additional experiment showing that MLPs cannot context scale even for classification. Below, we consider a classification problem where the label is given by the “sign” of the output of a linear regression model - if the sign is positive, the label is 0 and if it is negative, the label is 1. We find MLP performance first improves and then gets worse with increasing context length indicating a failure to context scale.
> >
> > Not meaning to split hairs, but your setup sounds more like logistic regression than classification. By classification, I had more in mind a setup with multiple clusters defined in-context, as in [Reddy 2024](https://arxiv.org/abs/2312.03002), which Tong and Pehlevan seem to show *does* convincingly context scale, even by your definition. Do you have an idea why this might be the case?
> >
> > > Note that by context-scaling, we refer to a setting in which the number of pretraining tasks is fixed, and we expect performance to improve as context examples increase – rather than initially improve and then degrade at a certain point.
> >
> > I'm not sure if I buy this perspective. If we take the numbers you provide in your second table and truncate them at context length 80 (a relatively long context, compared to where you start), it would appear that all models (including the MLP) context-scale quite convincingly. At least in this fairly large window, the MLPs performance improves substantially with longer context. That the MLPs' performance subsequently declines seems to be an independent issue. It's certainly up to you how you define the notion of "context scaling" exactly, but it seems somewhat disingenuous to claim that the MLP does not context scale when its performance improves substantially with longer contexts.
> >
> > > The derivation is in Appendix B
> >
> > My apologies, there doesn't seem to be any derivation regarding the Hilbert estimate and Transformers in Appendix B. Perhaps it's still elsewhere, or I'm missing something blatant? I'm certain my math skills are not as sharp as yours, so I would love to see this worked out in detail, even if it's obvious!
> >
> > Thanks again for the notes. Unfortunately my concerns remain, and I maintain my current score.

---

> > > ### Author Response · Authors · 2024-11-26
> > >
> > > > **Reviewer comment:** Not meaning to split hairs, but your setup sounds more like logistic regression than classification. By classification, I had more in mind a setup with multiple clusters defined in-context, as in Reddy 2024, which Tong and Pehlevan seem to show does convincingly context scale, even by your definition. Do you have an idea why this might be the case?
> > >
> > >
> > > **Our response :**  We would like to clarify that our paper is not primarily focused on classification; rather, we included a classification task as an example to address your concerns. Furthermore, regarding Tong and Pehlevan, could you please specify which part of their paper you are referring to? In Figure 1, they vary the context length for a classification task, but we do not find any evidence of context scaling in their results.
> > >
> > > > **Reviewer comment:** ... At least in this fairly large window, the MLPs performance improves substantially with longer context. That the MLPs' performance subsequently declines seems to be an independent issue. It's certainly up to you how you define the notion of "context scaling" exactly, but it seems somewhat disingenuous to claim that the MLP does not context scale when its performance improves substantially with longer contexts.
> > >
> > >
> > > **Our response :**  We would like to point out that Figure 7 includes four experiments. In three of them, the MLP clearly does not exhibit task-scaling behavior at all. In the specific case you mention (top-right panel), while there is an initial improvement with increasing context length, the performance does not continue to improve with further increase of in-context length, in contrast to attention based features \psi_l and \psi_H that provides consistent improvement with increased context length.
> > >
> > >
> > > > **Reviewer comment:**  ... there doesn't seem to be any derivation regarding the Hilbert estimate and Transformers in Appendix B...
> > >
> > > **Our response :** Please see lines 756-773. This derivation holds for any choice of kernel including the Hilbert kernel. Please let us know if you would like any further clarification.

---

> > > > ### Comment · Reviewer_NaCy · 2024-11-26
> > > >
> > > > Thanks for the additional clarifications!
> > > >
> > > > >We would like to point out that Figure 7 includes four experiments. In three of them, the MLP clearly does not exhibit task-scaling behavior at all. In the specific case you mention (top-right panel), while there is an initial improvement with increasing context length, the performance does not continue to improve with further increase of in-context length, in contrast to attention based features \psi_l and \psi_H that provides consistent improvement with increased context length.
> > > >
> > > > This is certainly true, so perhaps your claim should be adjusted to "MLPs sometimes context scale, and sometimes do not context scale," or even "MLPs context scale for small context lengths given sufficient data"? To claim that "MLPs do not context scale" seems incorrect (or at least, overly coarse) in light of this evidence.
> > > >
> > > > >Please see lines 756-773. This derivation holds for any choice of kernel including the Hilbert kernel.
> > > >
> > > > Perhaps my confusion is the following. It seems like there are two possible interpretations to your claim about the Hilbert estimate:
> > > > 1. A Transformer's self-attention can implement the Hilbert estimate.
> > > > 2. A Transformer's self-attention matrix can be interpreted as a kernel smoothing operator. One such kernel is the Hilbert estimate.
> > > >
> > > > I interpreted your claim to be (1), but it sounds like you actually intend (2)? If the latter, I'm unsure how Hilbert estimates fit into your overall argument, and why you included them? Why not stop at the 1-step GD kernel (in eq 10), which has a natural connection to the attention matrix structure? Would a Transformer implement anything that looks like a Hilbert estimate in practice?

---

> > > > > ### Author Response · Authors · 2024-11-28
> > > > >
> > > > > Thank you for response.
> > > > >
> > > > > > **Reviewer comment:** This is certainly true, so perhaps your claim should be adjusted to "MLPs sometimes context scale, and sometimes do not context scale," or even "MLPs context scale for small context lengths given sufficient data"? To claim that "MLPs do not context scale" seems incorrect (or at least, overly coarse) in light of this evidence.
> > > > >
> > > > > **Our response :** When we state that a model "context scales," we mean it consistently improves as more context information is provided. Consistency is crucial because, with additional context data, one naturally expects a model to perform better. We will clarify this in the final version of the paper. Specifically, we emphasize that while MLPs may initially show improvements with added context, they do not consistently scale with context length.
> > > > >
> > > > > > **Reviewer comment:** Perhaps my confusion is the following. It seems like there are two possible interpretations to your claim about the Hilbert estimate:
> > > > > >1. A Transformer's self-attention can implement the Hilbert estimate.
> > > > > >2. A Transformer's self-attention matrix can be interpreted as a kernel smoothing operator. One such kernel is the Hilbert estimate.
> > > > >
> > > > > >I interpreted your claim to be (1), but it sounds like you actually intend (2)? If the latter, I'm unsure how Hilbert estimates fit into your overall argument, and why you included them? Why not stop at the 1-step GD kernel (in eq 10), which has a natural connection to the attention matrix structure? Would a Transformer implement anything that looks like a Hilbert estimate in practice?
> > > > >
> > > > > **Our response :**  Both interpretations (1) and (2) are correct, and we appreciate the opportunity to clarify. As mentioned in line 431, if the kernel is chosen to be the exponential kernel, then \psi_K implements the soft-max attention head, which aligns with the original GPT-based attention head. However, we highlight the Hilbert kernel because the exponential kernel smoother is not a consistent predictor—it does not converge to the optimal solution. This distinction is why we emphasize the Hilbert kernel in our argument, as it provides a more robust theoretical foundation for understanding self-attention's capabilities and gives consistency guarantee for any statistical task.

---

> > > > > > ### Comment · Reviewer_NaCy · 2024-12-01
> > > > > >
> > > > > > Thanks for the additional clarification.
> > > > > >
> > > > > > >When we state that a model "context scales," we mean it consistently improves as more context information is provided. Consistency is crucial because, with additional context data, one naturally expects a model to perform better. We will clarify this in the final version of the paper. Specifically, we emphasize that while MLPs may initially show improvements with added context, they do not consistently scale with context length.
> > > > > >
> > > > > > This is fine, and makes for an interesting take, but should be clarified in your manuscript. Reading your document, it was not clear to me what you meant by "MLPs do not context scale" when their performance appears to improve sometimes as the context length increases. If you mean something weaker, this should be stated in the text.
> > > > > >
> > > > > > >Both interpretations (1) and (2) are correct, and we appreciate the opportunity to clarify. As mentioned in line 431, if the kernel is chosen to be the exponential kernel, then \psi_K implements the soft-max attention head, which aligns with the original GPT-based attention head. However, we highlight the Hilbert kernel because the exponential kernel smoother is not a consistent predictor—it does not converge to the optimal solution. This distinction is why we emphasize the Hilbert kernel in our argument, as it provides a more robust theoretical foundation for understanding self-attention's capabilities and gives consistency guarantee for any statistical task.
> > > > > >
> > > > > > My apologies, I still don't understand. If (1) is correct -- that is, a Transformer (in the original sense, with softmax attention) can implement the Hilbert estimate -- how do you show this? Is there a derivation somewhere in the manuscript? You appear to claim that softmax attention implements the exponential kernel, which you state is not a consistent predictor (whereas a Hilbert estimate is)? Do you mean to imply that a different choice of attention nonlinearity will implement a Hilbert estimate instead?

---

> > > > > > > ### Author Response · Authors · 2024-12-02
> > > > > > >
> > > > > > > > **Reviewer comment:** This is fine, and makes for an interesting take, but should be clarified in your manuscript...
> > > > > > >
> > > > > > > **Our response :**  We are happy to clarify this point in the final manuscript. We will emphasize that while MLPs may initially show improvements with added context, they do not consistently scale with context length.
> > > > > > >
> > > > > > > > **Reviewer comment:** My apologies, I still don’t understand. If (1) is correct -- that is, a Transformer (in the original sense, with softmax attention) can implement the Hilbert estimate -- how do you show this? Is there a derivation somewhere in the manuscript? You appear to claim that softmax attention implements the exponential kernel, which you state is not a consistent predictor (whereas a Hilbert estimate is)?
> > > > > > >
> > > > > > > **Our response:** To clarify, softmax attention does not implement the Hilbert estimate.  Instead, softmax attention implements a kernel smoother with the exponential kernel.  Indeed, this has been shown in prior work [see, e.g., Transformer Dissection: A Unified Understanding of Transformer’s Attention via the Lens of Kernel](https://arxiv.org/abs/1908.11775).
> > > > > > >
> > > > > > > > **Reviewer comment:**   Do you mean to imply that a different choice of attention nonlinearity will implement a Hilbert estimate instead?
> > > > > > >
> > > > > > > **Our response :**  Yes, provided the data is on the unit sphere in d dimensions, attention can implement the Hilbert estimate simply by changing the nonlinearity.  For data not on the unit sphere(general case), we implemented the Hilbert estimate directly by replacing attention.

---

> > > > > > > > ### Comment · Reviewer_NaCy · 2024-12-03
> > > > > > > >
> > > > > > > > Thanks for the additional notes, and engaging in a long discussion!
> > > > > > > >
> > > > > > > > > To clarify, softmax attention does not implement the Hilbert estimate. Instead, softmax attention implements a kernel smoother with the exponential kernel.
> > > > > > > >
> > > > > > > > This part of the discussion still confuses me, and I remain unsure about the overall intent. Is the goal to understand something about Transformers? If so, why bother with the Hilbert estimate in the first place, given that a vanilla Transformer (with softmax attention) cannot implement it? Or is it to demonstrate that kernel methods (including a kernel-based interpretation of attention) can context-scale? Though you weren't able to show consistency for the exponential kernel, why does it still seem to context scale quite well in your experiments? Put another way, I'm unsure how the theory you describe fits into the broader picture of your results, and it remains unclear if it's theory for the sake of having a theory component to your manuscript, or whether you're hoping to describe something deeper?
> > > > > > > >
> > > > > > > > If you're able to clarify your point about context scaling in MLPs, and refine your theoretical treatment of this subject, I think this would make for a fantastic paper. Given the current issues, however, I retain some reservations about recommending full acceptance. I update my score to a 6.

---

> > > > > > > > > ### Author Response · Authors · 2024-12-03
> > > > > > > > >
> > > > > > > > > We sincerely thank the reviewer for engaging in an extended discussion. This provides us with an excellent opportunity to clarify  our work.
> > > > > > > > >
> > > > > > > > > **Regarding context scaling in MLPs:** We will explicitly clarify this point in the final manuscript that while MLPs may initially exhibit improvements with added context, they do not consistently improve with increasing context length.
> > > > > > > > >
> > > > > > > > > **On the topic of the Hilbert kernel and its relationship to softmax attention:**  Our theoretical results aim to illustrate that standard softmax attention is a specific instance of a more general and powerful algorithm—kernel smoothing. This perspective  reveals that kernel smoothing with Hilbert kernel can be consistent even without requiring additional learned parameters. By framing softmax attention within this broader kernel-based perspective, we hope to inspire the development of more general attention mechanisms in future works.

---

### Official Review · Reviewer_2MBm · 2024-11-02

**Soundness:** 3
**Presentation:** 3
**Contribution:** 3
**Rating:** 6
**Confidence:** 3

**Summary:**

This paper studies context-scaling and task-scaling of ICL, under multiple ICL regression tasks, including linear regression with fixed and multi noise level, two-layer ReLU neural networks, decision trees, and sparse linear regression. Experiments are conducted on GPT2 and Simplified GPT (SGPT) by taking key, query and value matrices to be identify matrices and removing batch normalization. Under those tasks, GPT2 and SGPT both demonstrate context-scaling ability and the performances of GPT2 and SGPT are very close to each other. SGPT allows a kernel smoothing perspective interpretation of transformer's ICL capability. Specifically, the authors demonstrate the capability of ICL capability of transformer by showing a single layer SGPT can perform kernel smoothing (including the consistent Hilbert estimate as special case) with appropriate feature map corresponding to the attention. To see what statistics are the essence of task-scaling and context-scaling, experiments on MLP with different inputs , such as vectorized input data with or without kernelized features are conducted. Specifically, task-scaling attributes to the vectorized data and context scaling attributes to the kernelized features, and combining both inputs provide both task-scaling and context scaling.

**Strengths:**

The SGPT considered and its experiments are novel. And it is quite surprising and interesting to see that its performance is comparable to GPT2. I suppose it is due to the simplicity of the ICL tasks conducted in the paper.

The idea of connecting ICL and kernel smoothly is clearly presented and is of insight.

The separation of context-scaling and task-scaling via feature from kernel estimate and vectorized input is novel and can potentially help us understand their impacts better individually.

**Weaknesses:**

Though with the consistency of Hilbert estimate, how exactly the transformer performs Hilbert estimate i.e., via the construction of activation function in attention, is not straightforward.

What is the major intuition of taking key, query, and value matrices to be identity? Such intuition is vital since it is shown that the context-scaling capability is attributed to the attention, and task-scaling is to the MLP with vectorized data. I wonder if key, query, and value matrices are learnable, will they also provide sufficient task-scaling ability?

What is the theoretical or explanatory justification for the capability of the task-scaling ability of transformer and MLP, generalization?

**Questions:**

What is the task-scaling performance for single-layer SGPT (a task-scaling counterpart of Fig 5)?

Could you explain more on the right panel of Fig 3?

Could you explain more on the Fig 4(B), 2-layer NN, SGD?

---

> ### Author Response · Authors · 2024-11-18
>
> We thank the reviewer for the positive review. We will address the questions as following,
>
> > **Reviewer comment:**
> > What is the major intuition of taking key, query, and value matrices to be identity? Such intuition is vital since it is shown that the context-scaling capability is attributed to the attention, and task-scaling is to the MLP with vectorized data. I wonder if key, query, and value matrices are learnable, will they also provide sufficient task-scaling ability?
>
> **Our response :**  The main reason for setting these matrices to be the identity is that it drastically simplifies the model while still being competitive with  GPT-2.  Furthermore, the fact that our model can both context and task-scale shows that key, query, value weight matrices are not necessary for models to exhibit  these properties.
>
> > **Reviewer comment:**  What is the theoretical or explanatory justification for the capability of the task-scaling ability of transformer and MLP, generalization?
>
> **Our response :**  In the case of a fixed context length, the input to the MLP is a combination of the raw data and an estimation of the label. Consequently, the MLP can be viewed as acting like a comparison model across tasks—somewhat analogous to a weighted k-nearest neighbor model. However, developing a rigorous theoretical foundation for this capability is beyond the scope of the current paper.
>
> > **Reviewer comment:**  What is the task-scaling performance for single-layer SGPT (a task-scaling counterpart of Fig 5)?
>
> **Our response :** Here’s the performance of 1-layer SGPT:
>
> ### Task: Linear Regression
>
> | # Pretraining Tasks | 1k   | 10k   | 100k  | 1M   | 5M   |
> |----------------------|-------|-------|-------|-------|-------|
> | Context Length 10    | 0.545 | 0.545 | 0.540 | 0.476 | 0.470 |
> | Context Length 40    | 0.292 | 0.290 | 0.285 | 0.217 | 0.210 |
>
> ### Task: 2-Layer Neural Network
>
> | # Pretraining Tasks | 1k   | 10k   | 100k  | 1M   | 5M   |
> |----------------------|-------|-------|-------|-------|-------|
> | Context Length 10    | 0.864 | 0.860 | 0.846 | 0.841 | 0.830 |
> | Context Length 40    | 0.683 | 0.675 | 0.670 | 0.663 | 0.650 |
>
>
>
> > **Reviewer comment:** Could you explain more on the right panel of Fig 3?
>
> **Our response :** This plot zooms in on a fixed context length of 30 and shows that for both noise levels, the performance is comparable to ridge regression corresponding to each noise level. The black dots represent different ridge values, with the x-axis showing the MSE for the first noise level and the y-axis showing the MSE for the second noise level. If we were to choose a fixed ridge value, it’s clear that each noise level would require a different value, and no single value would work well for both noise levels. This plot demonstrates that both GPT-2 (also previously shown in Bai et al. (2023)) and SGPT (ours) can perform well across both noise levels as good as ridge regression. We will add this explanation to the Appendix.
>
>
>
> > **Reviewer comment:** Could you explain more on the Fig 4(B), 2-layer NN, SGD?
>
> **Our response :** This is the baseline previously used in Garg et al. (2023), where a similar 2-layer neural network architecture with new random initialization is considered and fine-tuned on the context data points. We will add this explanation to the Appendix.
>
> Thank you again for your positive feedback and thoughtful evaluation. We appreciate your support and are happy to address any additional points if needed.

---

### Official Review · Reviewer_thSw · 2024-11-04

**Soundness:** 2
**Presentation:** 2
**Contribution:** 2
**Rating:** 3
**Confidence:** 3

**Summary:**

The authors study an important question: how does in-context learning in Transformers depend on the number of in-context examples as well as the number of overall tasks? They draw a connection to kernel smoothing and demonstrate that a simplified version of the Transformer architecture can implement this algorithm equally well. They show that in contrast to Transformers, MLPs do not exhibit scaling with in-context examples, but that a feature map inspired the Transformer's mechanism can yield successful improvement with more in-context examples.

**Strengths:**

- I really liked the introduction and thought that the question of context-scaling was nicely set up.
- I appreciated the authors considering a wide range of different tasks that they collected from the relevant literature.
- Building part of the SGPT into the MLP was a neat method for illustrating the origin of the distinct mechanism between either.

**Weaknesses:**

Unfortunately, I do not think this work is ready for publication in its current state. Primarily, I believe that the paper does not provide sufficiently novel insight from the prior literature.

Notably, improvement of Transformer performance with the number of in-context examples was already noted (as the authors lay out in the related work section), e.g. in Bai et al. (2023) and the prior theoretical literature also explains why this would be the case, as it draws the connection between ICL in Transformers and gradient descent and kernel smoothing --- both of which improve with the number of samples. I do not understand why previous theoretical results (e.g. Proposition 1 in von Oswald et al. (2023)) would only apply to fixed context length.

It is also unclear to me how SGPT provides a novel insight into the mechanism of ICL compared to, say, the construction in von Oswald et al. (2023), who also explicitly draw the connection to kernel smoothing and use a similar set of simple key, query, and value matrices (especially when $W_0=0$ in their construction in Appendix A.1). The authors argue that the simplicity of SGPT is a substantial strength of this paper, as it demonstrates that there are many problems such a simple architecture can solve. But it is unclear to me whether this a theoretical argument (which seems to make the connection to kernel smoothing, in which case I'm unsure how this is different from the insight by von Oswald et al.) or an empirical argument (in which case I think the authors would have to demonstrates concretely that SGPT outperforms kernel smoothing algorithms).

As I noted, I think the contrast to MLPs and providing the modified features to the MLPs was interesting. I think it would be important, however, to provide insight into *how* the vectorized component enables them to scale with the number of examples.

Taken together, I think the paper in its current form is not sufficiently distinct from existing work --- or at least does not explain sufficiently clearly how it is different. As I noted above, I do think that the authors focus on a really interesting question (context scaling) that provides a different angle from prior work. However, I think for the paper to be ready for publication, I think this investigation would have to further explore how this angle can change our theoretical understanding of context scaling.

**Questions:**

- Why do previous theoretical results only apply to fixed context length, as stated in l.157-159? (See weaknesses.)
- The SGPT as defined in Eq. 13 appears to be more specific than the generic feature map $\psi$ you are then introducing in Equation (9). Is that true and if so, can you explain how this generalized version connects to the SGPT as well as the generic Transformer architecture?
- Tong & Pehlevan (2024) also show that MLPs cannot context scale (Fig. 1d). This is currently not reflected in your related work section (l. 143-146). Could you please clarify and explain how your findings relate to these prior findings?

Minor comments:
L. 51: “an non-exhaustive” -> “a non-exhaustive”
L. 157: typo?
L. 213: “identify” -> “identity”

---

> ### Author Response · Authors · 2024-11-18
>
> We thank the reviewer for the comments. We address the concerns as following,
>
> > **Reviewer comment:**
> > Notably, improvement of Transformer performance with the number of in-context examples was already noted (as the authors lay out in the related work section), ... I do not understand why previous theoretical results (e.g. Proposition 1 in von Oswald et al. (2023)) would only apply to fixed context length.
>
> >It is also unclear to me how SGPT provides a novel insight into the mechanism of ICL compared to, say, the construction in von Oswald et al. (2023), who also explicitly draw the connection to kernel smoothing and use a similar set of simple key, query, and value matrices (especially when W0=0 in their construction in Appendix A.1). .... But it is unclear to me whether this a theoretical argument (which seems to make the connection to kernel smoothing, in which case I'm unsure how this is different from the insight by von Oswald et al.) or an empirical argument (in which case I think the authors would have to demonstrates concretely that SGPT outperforms kernel smoothing algorithms).
>
>
> **Our response :**
> A novel insight of our work beyond that of von Oswald et al. (2023) is that SGPT improves over both standard kernel smoothers and 1-step of GD.  Indeed, one-layer of SGPT is better than both of these previous algorithms as it combines an MLP and kernel smoothing to simultaneously task and context scale.  Empirically, we demonstrate in Fig. 7 that combining MLP with kernel smoothing features significantly outperforms 1-step of GD and kernel smoother on both linear regression and nonlinear tasks.  This contrasts with von Oswald et al. (2023), who focus on demonstrating the equivalence of a one-layer transformer to a single step of GD (or kernel smoothing in their Appendix A.1).
>
>
> > **Reviewer question:**
> >Why do previous theoretical results only apply to fixed context length, as stated in l.157-159? (See weaknesses.)
>
> **Our response :**
> A fixed context length is part of the assumptions in the previous theoretical results (Von Oswald et al. (2023), Ahn et al. (2023), Zhang et al. (2024a), Mahankali et al. (2024), Zhang et al. (2024b)). In particular, Proposition 1 in Von Oswald et al. (2023) states that for any given $N$ pairs of context examples, there exists a transformer such that its output is identical to the one-step GD output over the $N$ pairs of context examples. Therefore, Proposition 1 cannot be used if we test the same transformer with a different number of context examples.
>
> The only work we are aware of for analyzing varying context-length is Theorem 5.3 in Wu et al. (2024), which allows testing a pre-trained attention model with a varying context length. However, their results are only tight when the context length is close to the one used in training.
>
>
> > **Reviewer question:**
> >The SGPT as defined in Eq. 13 appears to be more specific than the generic feature map \psi you are then introducing in Equation (9). Is that true and if so, can you explain how this generalized version connects to the SGPT as well as the generic Transformer architecture?
>
> **Our response :**
> Thank you for pointing this out. Equations 13 and 9 are essentially the same – the only difference is the second residual connection in the transformer. In the analysis presented in Section 5, we observed that adding this residual did not significantly impact the performance across the experiments in that section. Therefore, we chose to omit it in Equation 9 for the sake of simplicity.
>
>
> > **Reviewer question:**
> >Tong & Pehlevan (2024) also show that MLPs cannot context scale (Fig. 1d). This is currently not reflected in your related work section (l. 143-146). Could you please clarify and explain how your findings relate to these prior findings?
>
> **Our response :**
> Fig. 1d of Tong & Pehlevan is measuring excess risk, which refers to the difference between the model’s MSE and that of optimal ridge regression.  Optimal ridge regression varies with context length, thus it is possible for excess risk to increase even though MSE decreases. In contrast, our notion of context-scaling is defined based on raw MSE. Furthermore, in that figure, even transformers have higher excess risk (error bars indicate performance approaches that of random prediction) as context length increases even though we know transformers do context-scale.
>
> **Our response to reviewer minor comments:**
> Thank you for the comments. We will make the modifications.
>
> We hope our responses address your concerns. If any points remain unclear, we’d be happy to clarify. If the manuscript now aligns with your expectations, we’d appreciate your consideration of an updated score.

---

> > ### Comment · Reviewer_thSw · 2024-11-26
> >
> > Thank you for your response. Unfortunately, I do not think they address my fundamental concerns about the novelty of this work.
> >
> > > A novel insight of our work beyond that of von Oswald et al. (2023) is that SGPT improves over both standard kernel smoothers and 1-step of GD. Indeed, one-layer of SGPT is better than both of these previous algorithms as it combines an MLP and kernel smoothing to simultaneously task and context scale. Empirically, we demonstrate in Fig. 7 that combining MLP with kernel smoothing features significantly outperforms 1-step of GD and kernel smoother on both linear regression and nonlinear tasks. This contrasts with von Oswald et al. (2023), who focus on demonstrating the equivalence of a one-layer transformer to a single step of GD (or kernel smoothing in their Appendix A.1).
> >
> > I think this is a neat finding. However, it seems that your theoretical construction cannot account for this, correct? I think this could be a good motivation better understanding how exactly this MLP overcomes this prior theoretical constructs. However, without such deeper analysis, I do not think it is a sufficient result.
> >
> > > A fixed context length is part of the assumptions in the previous theoretical results (Von Oswald et al. (2023), Ahn et al. (2023), Zhang et al. (2024a), Mahankali et al. (2024), Zhang et al. (2024b)). In particular, Proposition 1 in Von Oswald et al. (2023) states that for any given pairs of context examples, there exists a transformer such that its output is identical to the one-step GD output over the $N$ pairs of context examples. Therefore, Proposition 1 cannot be used if we test the same transformer with a different number of context examples.
> >
> > > The only work we are aware of for analyzing varying context-length is Theorem 5.3 in Wu et al. (2024), which allows testing a pre-trained attention model with a varying context length. However, their results are only tight when the context length is close to the one used in training.
> >
> > Isn't your construction here the exact same used in von Oswald et al., except without the scaled projection matrix though? Please clarify if I'm misunderstanding.

---

> > > ### Author Response · Authors · 2024-11-28
> > >
> > > > **Reviewer comment:**   I think this is a neat finding. However, it seems that your theoretical construction cannot account for this, correct? I think this could be a good motivation better understanding how exactly this MLP overcomes this prior theoretical constructs. However, without such deeper analysis, I do not think it is a sufficient result.
> > >
> > > **Our response :**  We are not entirely sure what you mean by "theoretical construction," since we do not have any weight construction in our work. we appreciate a clarification.
> > >
> > > However, our explanation does indeed provide insight into why the MLP enables SGPT to outperform the prior understanding that transformers merely implement a single step of gradient descent. Specifically, our work shows that while attention effectively implements an in-context estimator (e.g., a single step of gradient descent or kernel smoothing), the MLP leverages additional information across tasks. A helpful way to conceptualize this is to view the MLP as acting like a 𝑘-nearest neighbors estimator across tasks, enriching the model's ability to generalize.
> > >
> > > Developing a rigorous theoretical framework to formalize this insight and analyze the corresponding sample complexity (of task-scaling) of MLPs would indeed be an interesting and valuable direction for future work. However, such an analysis is beyond the scope of this paper. Instead, we aim to provide the key intuition and empirical evidence to motivate further study in this direction.
> > >
> > > > **Reviewer comment:**  Isn't your construction here the exact same used in von Oswald et al., except without the scaled projection matrix though? Please clarify if I'm misunderstanding.
> > >
> > > **Our response :**  We want to emphasize again that we do not have any construction in our paper. Furthermore, If you are referring to Proposition 1 in von Oswald et al.'s work, their construction is limited to a fixed context length N_y(using their notation). While this is effective for their analysis, it does not account for how context scaling works, which is a key focus of our work. Our contributions even go beyond this by exploring the interplay between context and task scaling, which their construction does not address.

---

> > > > ### Comment · Reviewer_thSw · 2024-12-03
> > > >
> > > > Thank you for your response. What I mean by construction is your specification of $\psi$ in Section 5.1 and Appendix B.
> > > >
> > > > > We want to emphasize again that we do not have any construction in our paper. Furthermore, If you are referring to Proposition 1 in von Oswald et al.'s work, their construction is limited to a fixed context length N_y(using their notation). While this is effective for their analysis, it does not account for how context scaling works, which is a key focus of our work.
> > > >
> > > > I assume you mean the context length $N$, not $N_y$, right? I think $N_y$ in their paper is the output dimensionality. With that in mind, I think equation (8) in their paper could directly be applied to different $N$ and implement a gradient step, correct? The only distinction lies in the projection matrix $\eta/N I$; while this projection matrix would have to be held fixed at some value, this could be understood as different learning rates for different numbers of examples. As such, it is unclear to me why this previous result only applies to a fixed context length.
> > > >
> > > > It is true that this paper does not explicitly point out an application to different context length and I want to be clear that I think there is value in explicitly studying the interplay between context scaling and task scaling. I'm merely trying to better understand the claims about novel theoretical contributions in this paper.

---

> > > > > ### Author Response · Authors · 2024-12-03
> > > > >
> > > > > Thank you for your response.
> > > > >
> > > > > > **Reviewer comment:** What I mean by construction is your specification of in Section 5.1 and Appendix B.
> > > > >
> > > > > **Our response :** The \psi_L feature map in section 5.1 come from the prior work as we have cited them. We did not claim construction these features are part of our contribution. Our contribution is to show these features, when used together with raw features, can achieve context-scaling empirically.
> > > > >
> > > > > > **Reviewer comment:** With that in mind, I think equation (8) in their paper could directly be applied to different and implement a gradient step, correct?As such, it is unclear to me why this previous result only applies to a fixed context length.
> > > > >
> > > > > **Our response :** In the construction of Proposition 1, although we can apply equation (8) to different context lengths, equation (8) does not provide any performance guarantee as the "learning rate" might be suboptimal. Therefore, Proposition 1 does not address context-scaling.
> > > > >
> > > > > > **Reviewer comment:** I'm merely trying to better understand the claims about novel theoretical contributions in this paper.
> > > > >
> > > > > **Our response :** As we mentioned in lines 118 to 122, our theoretical contribution is by viewing attention form kernel smoothing perspective. Specifically, when the Hilbert estimate is chosen as the smoothing method, the model implements a statistically optimal (consistent) estimate as the context length approaches infinity.
> > > > >
> > > > > Our main contribution of our paper is as mentioned from line 114-125.

---

### Meta-Review · Area_Chair_63Ze · 2024-12-22

**Metareview:**

This paper explores task and context scaling in in-context learning. The authors propose simplified models like SGPT and make theoretical connections to kernel smoothers. While the ICL topic is very timely and relevant, the reviewers have concerns regarding the novelty and depth of insights. The claims regarding SGPT’s advancements over prior works are not convincingly justified, and some of the findings seem incremental. Reviewers found certain theoretical contributions to be unclear and there was also a concern on insufficient experimentation to substantiate claims. For example, SGPT matching GPT-2's performance lacks thorough ablations. Furthermore, the analysis/discussion of MLPs raises valid questions that remain not fully unresolved. Given the limited novelty and incomplete discussions, I recommend rejecting this submission in its current form.

**Additional Comments On Reviewer Discussion:**

During the rebuttal, reviewers highlighted concerns about the lack of clarity and novelty in contributions. Despite authors' responses, the explanations failed to address key issues, such as the theoretical novelty of SGPT and the ambiguous role of attention versus MLPs in in-context learning. Reviewers like thSw found the explanations for theoretical extensions over von Oswald et al. insufficient. Here, I also agree that kernel smoothing viewpoint is likely not novel and discussed by prior art (at least implicitly) including by von Oswald et al., Collins et al., Chen et al. and Nguyen et al. (FourierFormer, NeurIPS'22, not cited). Similarly, Reviewes NaCy and DfAQ remained unconvinced by the experimental rigor and theoretical integration. In summary, while the paper makes insightful points and has good potential, it currently needs further revision and improvement for acceptance.

---

### Decision · Program_Chairs · 2025-01-22

Reject